# "Pet effect" patterns: Dynamics of animal presence and caregiver affect across (tele) work and non-work contexts

**Joni Delanoeije**[1,2,3]*, **Miriam Engels**[4], **Mayke Janssens**[4,5]

**1** Work and Organisation Studies, Faculty of Economics and Business, KU Leuven, Leuven, Belgium,
**2** Research Foundation Flanders (FWO), Brussels, Belgium, **3** International Association of Human-Animal Interaction Organizations (IAHAIO), Seattle, Washington, United States, **4** Faculty of Psychology, Open University, Heerlen, Netherlands, **5** Department of Psychiatry and Psychology, School for Mental Health and Neuroscience, Maastricht University Medical Centre, Maastricht, Netherlands

* joni.delanoeije@kuleuven.be

## Abstract

Human-animal interactions (HAI) may relate to animal caregivers' affect, also referred to as the "pet effect". However, studies have not explored these associations in work contexts or evaluated longitudinal patterns of HAI with other activities across work and non-work contexts, and their associations with caregiver emotions. We therefore assess momentary associations between HAI during (tele)work and non-work time and positive and negative caregiver affect (PA/NA), identify person-level patterns of longitudinal HAI state trajectories, and analyze cross-level moderating effects of these patterns on momentary associations between HAI and PA/NA. First, we evaluated associations between momentary HAI and caregiver PA/NA including the moderating role of momentary work state (teleworking vs. not working). Second, using a data-driven approach, we applied sequence analysis to determine heterogeneity in state trajectories amongst caregivers using working activity and animal presence in five possible states (working at work/teleworking with animal/ teleworking without animal/not working with animal/not working without animal), which we labelled as animal-work constellations. Similar trajectories of animal-work constellations across caregivers were grouped into clusters with recognizable patterns. Third, we assessed whether such patterns moderated momentary associations between HAI and caregiver PA/NA. Caregivers ($N_{persons}$ = 324) completed ecological momentary survey data during five days with 10 prompts per day ($N_{observations}$ = 16,127) between 2017 and 2024. Results showed that momentary associations between HAI and affect were moderated by momentary work state and person-level animal-work constellation pattern, contextualizing the "pet effect". Our results highlight the importance of microlevel investigations of animal-work constellations and validates the novel use of sequence analysis to explore the role of context and time.

**Data availability statement:** Data are available upon request for qualified researchers who meet specific criteria, subject to evaluation by the Research Ethics Committee of the Open University (cETO; OU; U2016/00165/CBO and U2022/08386). The contact person for data availability request is Professor Dr. Nele Jacobs, Department Head, Life Span Psychology, Open University, the Netherlands; e-mail address: nele.jacobs@ou.nl, who is responsible for data requests dispatched to their department and was not involved in the study. The collaborating researcher from the Open University (Professor Mayke Janssens), staff of the Open University, will be responsible for all data storage. Data will be stored and preserved for a minimum of 5 years on the internal storing system of the Open University. Data will be removed after this period with at maximum 10 years after publication. Data will not be shared with third parties. E-mail addresses will be kept separate from respondents' answers and individual data will only be reported at an aggregated level or by means of data suppression. We take all necessary legal precautions regarding data storage and preservation.

**Funding:** This work was supported by the Research Foundation Flanders (FWO-Vlaanderen; https://www.fwo.be) under Grant 12B0522N, which was awarded to JD. The funder did not play a role in the study design, data collection and analysis, decision to publish, or preparation of the manuscript.

**Competing interests:** The authors have declared that no competing interests exist.

## Introduction

Interactions with companion animals, or human-animal interactions (HAI), may enhance the mental health of the caregiver of the animal as supported by a multitude of studies documenting the beneficial relationship between HAI and human mental health outcomes, including stress [1,2] and affect [2,3] (for a meta-analysis, see [4]). The positive impact of HAI on caregivers' mental health, particularly regarding their affective well-being, is frequently referred to as the "pet effect". This term was initially introduced by Allen [5] and has since been adopted by various scholars (e.g., [3,6–9]). However, studies have also reported negative effects and hence, well-being effects of HAI are considered to be mixed (see [10] and [11] for systematic reviews and [12] for a meta-analysis). Explanations for observed inconsistencies include methodological issues that bias towards observing positive rather than negative impacts [13–15]. Additionally, cross-sectional study designs do not allow for capturing the necessary sensitivity in momentary fluctuating HAI, decreasing study validity and limiting process and outcome evaluation [15–17]. Finally, effects have been found to depend on contextual factors including caregivers' living environment [18], employment status [19] and performed activities in daily life [20], suggesting a moderating impact of people's daily life and working contexts. Indeed, human-animal relationships are complex and multifaceted, requiring the consideration of individual contextual differences to account for both within-person and between-person fluctuations—both positive and negative—in the impact of companion animals on caregiver mental health [21]. Ecological momentary assessment (EMA), also referred to as experience sampling methodology (ESM), is an increasingly used method for examining the microlevel dynamics of human well-being in daily life. This approach involves repeated measurements once or multiple times a day, allowing for the assessment of states as they occur in the moment or shortly thereafter (for a meta-analysis, see [22]). In this way, this methodology enhances methodological rigor, reduces recall bias [23], and increases the measurement validity of fluctuating constructs [24].

EMA has been applied to capture momentary associations between HAI and human mental health in people's private life [3,7,17,25] and (tele)working contexts [26–31]. Moreover, it has been extensively employed in assessing the impact of working environments on employee well-being [32–34], shedding light on significant distinctions between work and non-work settings [35–38]. This method facilitates the examination of crucial within-person fluctuations in people's environments and their experiences at home and at work [24,39].

Studies using EMA to examine HAI and human mental health indicate that the presence of animals may buffer against negative affect (NA) and increase positive affect (PA) in caregivers [3,7]. These positive associations between HAI and human affective well-being have been evidenced especially in dogs and, to a lesser extent, cats [20,40]. As such, experimental work has shown that interactions with dogs increase momentary caregiver PA [1,25] and that the degree of engagement with one's dog negatively predicts momentary caregiver NA [1]. Also findings from daily diary studies have suggested that daily interactions with one's dog while teleworking increase daily human positive affect [26,29], especially when caregivers report a strong relationship with their dog [26].

In addition to the use of EMA, scholars have been using methods that examine people's trajectories of momentary states over time and the identification of discernible, between-person varying patterns of such trajectories to better understand longitudinal mental health dynamics [41]. Specifically, momentary states across contexts over time form individual "state trajectories" which can be clustered in between-person varying patterns. That is, individuals exhibiting similar trajectory patterns are grouped into corresponding clusters. Identifying individual trajectories of HAI across contexts and over time is important as the *longitudinal organization* of momentary states rather than the states themselves may be explanatory for affective

well-being. The application of such methodology entails sequence analysis, a method that originated in biological research on DNA sequencing and has increasingly been used in other fields, including sociological studies demonstrating associations between people's employment trajectories and their well-being [42,43]. Hence, sequence analysis can be applied to different types of longitudinal data to describe differences between groups and contexts with a high temporal resolution. Though it has widely been used for the analysis of longer-term phenomena [42,43], it could also help zoom in on differences in daily activities or experiences [41].

We argue that this approach allows for contextualizing the "pet" effect in two main ways, improving our understanding of HAI and their effects. First, it allows for collecting *momentary data across daily life contexts* (e.g., at work, at home) of animals (e.g., animal presence) and caregivers (e.g., working activity). This is important since studies on HAI and affective well-being have mainly been limited to specific contexts, either focusing on caregivers' home context with little attention for their work or focusing on HAI in the workplace disregarding the home context. This is surprising given that the effects of HAI will likely differ based on the environment in which they occur [44,45], with important differences between teleworking and regular working environments [27]. Additionally, in the aftermath of the worldwide COVID-19 pandemic, many teleworking caregivers have been shifting part of their work to their home environment [46], and interactions with dogs during telework [26] and during working on site [28] have been found to impact employees' daily positive affect. Similarly, research on jobs involving human-animal collaboration to achieve work goals shows that interacting with animals impacts employee emotions [47]. These insights are likely relevant to (tele)work environments where animals are present. We therefore argue that it is important to capture various life domains when evaluating HAI and their effects, as they may take place across various locations (e.g., home and work) and activities (e.g., working or non-working).

Second, this approach allows for comprehensive assessment of how HAI states are *organized in relation to each other over time*. Specifically, it facilitates the capture of individual state trajectories and the examination of their longitudinal complexity. In doing so, features such as the order of states (e.g., working with an animal followed by spending time with an animal during non-working time) or within-trajectory variability (e.g., switching back and forth between states rather than uninterruptedly spending time in certain states) can be considered. This longitudinal complexity may nuance the relationship between HAI and caregiver affect. Specifically, based on longitudinal features, caregivers' state trajectories can be categorized into distinct clusters, each representing similar patterns of state trajectories among caregivers. These between-caregiver varying patterns may moderate the momentary associations between HAI and affect, indicating that the "pet effect" will depend on the organization of HAI states across contexts and over time.

Studies on the "pet effect" seem to have largely ignored the relevance of this longitudinal complexity. This is surprising since the way in which HAI unravel over time and alternate with other daily life events across contexts (e.g., going to work, which takes up a considerable amount of time in caregivers' lives) will likely impact the relationship between HAI and affective well-being. In fact, research has widely documented moderating impacts of various person-level variables on momentary HAI effects, such as respondent personality [48,49] or animal-caregiver relationship strength [26,50]. Likewise, between-caregiver varying patterns of HAI state trajectories may moderate momentary associations between HAI and mental health. As such, the "pet effect" may differ based on a caregiver's pattern of interaction dynamics. For instance, momentary associations between HAI and affect will likely differ between caregivers who spend long workdays without their animal and compensate with interacting with their animal after work hours, and teleworking caregivers who intermittently interact with their animal throughout the workday.

Therefore, the main objective of the current study is to gain a comprehensive understanding of when HAI in daily life contexts (i.e., across work and non-work locations and activities) relate to human affective well-being. We posit an approach that considers the within-person alternation and sequencing of states of "animal-work constellations" which encompass animal presence, working activity and location—resulting in individual state trajectories—and identify whether between-caregiver varying patterns of such trajectories exist. In this approach, we examine whether momentary work state influences the relation between HAI and affect. We do this by investigating the moderating role of work state on the association between the presence of a companion animal (present vs. absent) and affect (PA and NA). We then use sequence analysis to determine heterogeneity in animal-work constellations of momentary working activity (working or not working), location (at home or elsewhere) and animal presence (present or absent), resulting in six possible constellations (working at work with animal, working at work without animal, teleworking with animal, teleworking without animal, not working with animal, not working without animal). Next, we assess whether the within-person trajectories of these constellations can be clustered into distinct patterns that vary between individuals. Finally, we examine whether these patterns of constellations moderate the momentary associations between HAI and affect.

Our work contributes to the literature in two main ways. First, we include caregiver's momentary work location and activity to contextualize the momentary association between HAI and human affective well-being. In doing so, we address recent calls to consider the daily life context in which HAI occur when evaluating their effects [15,51–53]. Next to considering the human, the animal, and their interaction, we consider the daily life context, applying a comprehensive approach [27,54]. Second, we assess HAI over time and across contexts and analyze whether between-person patterns in their longitudinal organization moderates momentary effects. This approach contextualizes the "pet effect" by assessing boundary conditions for associations between HAI and human affect (i.e., understanding when and for whom HAI relate to affective well-being). Such an approach addresses recent calls for the need to contextualize HAI [15,27,54] by looking at individual boundary conditions to understand the contexts in which favorable or unfavorable effects occur. In doing so, we combine the assessment of momentary associations between HAI and affect with the identification of caregiver-level patterns explanatory for momentary effects.

## Materials and methods

### Procedure and sample

We utilized data from two EMA studies conducted between 2017 and 2024 [3,7,55]. Recruitment in the first study took place between 1 February 2017 and 1 December 2022; recruitment in the second study took place between 29 November 2022 and 30 June 2024. The protocol for one of these studies was preregistered [55]. These studies were aimed at investigating the momentary "pet effect" in daily life with a focus on stress, affect and animal characteristics. Respondents had to be 18 years or older, living with at least one dog or cat and have access to a smartphone. In total, 324 respondents (31.8% male, and 68.2% female) participated in the two original studies using the data on which our analyses are based, resulting in a dataset of 16,127 measurement occasions. This sample size is comparable to similar studies in this area (e.g., [27,55]).

Recruitment was conducted by graduate and undergraduate students from various regions across the country to enhance geographical sample representation. To maximize demographic diversity, students recruited respondents within their personal networks and through social media, veterinary practices, animal equipment stores, and associations focused on caregivers.

Student-recruited convenience samples are deemed appropriate for complex research designs, including EMA studies [56,57]. This approach facilitated sample diversity in terms of age, gender, employment status, educational level, number of children, marital status, living situation and animal species.

Respondents first filled out an online questionnaire gathering data on demographic characteristics and information concerning their companion animals. After completion they were instructed to install a mobile application on their smartphone, the Real Life Exp application, to collect momentary data. For five consecutive days, ten times a day, respondents received a notification on their smartphone asking them to fill in a short questionnaire. Notifications were administered at semi-random intervals between 7:30 AM and 10:30 PM, with a minimum interval of 30 minutes between each notification. Upon receipt of each notification, respondents were prompted to report their current affect, activities, location, and the prevailing social context. They were also asked to report the presence of their dog and/or cat. To minimize recall bias and enhance reliability, respondents were instructed to complete the questionnaire immediately after receiving the notification. The questionnaire remained accessible for up to 15 minutes before becoming unavailable.

All respondents provided written digital informed consent. The study was approved by the Research Ethics Committee (cETO) of the Open University of the Netherlands (OU; U2016/00165/CBO and U2022/08386).

**Measures.** This study includes two types of measures: cross-sectional measures derived from the initial survey completed by respondents prior to the commencement of the EMA sampling, and longitudinal EMA measures collected as momentary data at 50 semi-random time points over a span of five consecutive days. Demographic information was extracted from the initial questionnaire.

**Momentary affect.** Momentary affect was assessed using ten affect-related adjectives derived from the Positive And Negative Affect Schedule [58]. The scale is composed of items that showed high loadings on respectively positive and negative affect latent factors and sufficient within-person variability in previous EMA studies ([7],[59–61]). All items are scored on a 7-point Likert scale (1 = not at all, 7 = very). The *positive affect* scale comprised of the statements: I feel "cheerful", "content", "happy" and "enthusiastic" (Cronbach's alpha$_{(within)}$ = 0.83, Cronbach's $\alpha_{(aggregated)}$ = 0.90). The mean score on these items indicated the level of PA, with a higher score reflecting more positive affect. The *negative affect* scale comprised of the statements: I feel "insecure", "lonely", "anxious", "irritated", "sad", and "guilty" (Cronbach's $\alpha_{(within)}$ = 0.67, Cronbach's $\alpha_{(aggregated)}$ = 0.82). The mean score on these items indicated the level of NA, with higher scores reflecting more negative affect.

**Work state.** Momentary work context was assessed based on the current location and activity. Current location was assessed with the item "Where I am now". Respondents were able to choose between several categories (e.g., at work, at home, at someone else's place, outside, etc.) which were recoded into the dichotomous variable "At home" (yes/no). Current activity was assessed with the item "What I am doing". Respondents were asked to choose their activity from several options (e.g., relaxing, physical activity/sports, taking care of others, household chores, work, etc.), which were recoded into the dichotomous variable "Working" (yes/no). Work state was determined based on respondents' current location and activity, resulting in three possible work states: Teleworking (at home, working); Working at work (not at home, working); Non-working (not working).

**Animal presence.** The presence of one's companion animal(s) was assessed with the dichotomous item: "At this moment,my companion animal is present" (yes/no).

**Human presence.** The presence of other people was assessed with the dichotomous item: "At this moment, I am alone (other people)" (yes/no; reverse scored).

**Animal-work constellation.** To analyze HAI across contexts, we combined information about respondents' current location, activity and animal presence from every EMA measurement to create a unique "animal-work constellation" state. Therefore, work state and animal presence were combined into one "animal-work constellation" variable with the following options: (1) Telework with animal, (2) Telework without animal, (3a) At work with animal, (3b) At work without animal, (4) Non-work with animal, (5) Non-work without animal and (6) No information (if at least one of the other variables was missing). The occurrence of state 3a, where respondents were at work with their animal, was notably rare, with less than 2% of the sample exhibiting this state. Therefore, this state was regrouped together with state 3b into: (3) At work with or without animal. The sum of these combined animal-work states make up the individual sequences across all 50 measures (including missed beeps) and later serve as the basis for the cluster analysis.

**Caregiver demographic information.** Age was measured by asking respondents to fill out a number for their age. Gender was measured by asking respondents "What is your gender?", with answering options including male, female, other, and prefer not to disclose. As only male (0) and female (1) were selected by respondents, this variable was dichotomous. Educational level was measured by asking respondents their level of education. Options were summarized in three main categories: basic education (0; no education, primary education), secondary education (1; shorter vocational, shorter secondary education, pre-university education) and tertiary education (2; intermediate vocational education, longer professional education, university). Employment status was measured using the following answering options: unemployed, household, school/ education, regular part-time work (up to 32 hours), and regular full-time work (32 hours or more). Civil status was measured using the question "What is your current civil status" (alone; in a relationship not living together; married/ living together; divorced; widowed). Living situation was based on the question "What is your current living situation?" with the answering options: one person household, with parents(s), with partner/family, and other. A new dichotomous variable was constructed to indicate whether a respondents was living alone or not (yes/no). Finally, as the present study focusses on dog and cat caregivers, number and species of animal(s) was measured using the following two items: (1) "How many dogs do you have?", (2) "How many cats do you have?". Based on these questions, two new dichotomous variables were created that indicated whether respondents were caregivers of dog(s) (yes/no) or cat(s) (yes/no).

## Data and analysis

We analyzed the data in three steps. The first step included the state-level analysis of the association between animal presence and affect, moderated by momentary work state. The second step included cluster analysis based on the animal-work constellation sequences (i.e., sequence analysis). The third step included the momentary analysis of the relationship between animal presence and affect, moderated by person-level animal-work cluster. All calculations and figures are based on Stata 15 with the Sadi extension for sequence analysis [62] and R.

To avoid long spells of missing data in the longitudinal sequence analysis, which is critical for Step 2 of the analysis, we employed state imputation methods. Specifically, we used responses to the question "Since the last notification, was your companion animal present?" to impute missing states. When this question was answered following a missing state, we imputed the previous state by incorporating the retrospective animal presence information, assuming that the work and location information remained constant. These imputations addressed approximately 10% of the data gaps and were used for the calculations of the sequences. Subsequently, we excluded respondents who still had more than two-thirds of states with missing information (n = 111), resulting in a final sample of 324 caregivers.

To rule out confounding explanations based on between-person variables solely accounting for cluster types and the moderating effect in Step 3, we assessed differences between clusters with regard to employment status, educational level, gender, age, number of children, animal species, marital status and living situation. We corrected for these differences by adding them as covariates to the models in Step 1 and Step 3.

**1. Momentary analysis animal presence and affect, moderated by work state.** For the momentary analysis of Step 1, animal presence and work state served as the independent variable and the moderator respectively. Multilevel regression models were performed with PA (Model 1) and NA (Model 2) as the dependent variable, animal presence and work state as the independent variables, and human presence, age and gender as control variables. Work state was tested as a moderator of the association between animal presence and PA/NA by adding the interaction term of animal presence and work state as a moderating independent variable. As there were insufficient notifications where the animal was present during 'working at work' states, this category was excluded from the analyses. Both models account for serial dependency allowing residuals to be correlated over time (AR(1)) and allow for intercepts and slopes to vary randomly across individuals.

**2. Cluster analysis/sequence analysis.** After combining animal presence, work state and location into one "animal-work constellation" state, we performed descriptive analyses on the sequences (see Table 1). This enabled us to describe patterns in terms of cumulative number of measurements in each state.

Next, we grouped individuals with similar patterns of animal-work constellations into clusters. In order to do this, we first calculated distances between each individual sequence and all others based on Optimal Matching (OM [63]). Hereby, distances between sequences are calculated based on the number of swaps necessary to make two sequences equal, either by substituting (so-called 'substitution costs') or by inserting and deleting states (so-called 'indel costs'). In contrast to the standard practice, where substitution costs are set to 1 and indel costs are set to 0.50, we created a customized distance matrix, in which the costs for a switch from work state to non-work state are considered more "expensive" than other changes, and the costs for substituting a missing state were set lower to avoid clustering based on missing information. Running OMA operations on the sequences resulted in a final distance matrix which we then used to group sequences into clusters with the help of cluster analysis (Ward's linkage). To determine the most appropriate number of clusters, we compared solutions between three and 10 clusters in terms of cluster size, content validity, and cluster solution (i.e., whether a higher solution adds another cluster of interest). We also calculated Duda-hart and Calinsky measures based on within- and between-errors from clusters with no clear results regarding the quality of the solution.

**3. Momentary analysis animal presence and affect, moderated by cluster.** In order to determine whether the momentary "pet effect" is dependent upon cluster, we performed multilevel random regression models with PA (Model 1) and NA (Model 2) as the dependent variable, animal presence and cluster as the independent variables, and—just as in Step 1—human presence, age and gender as control variables. Additionally, we added job type, animal species (dog, yes/no), marital status, educational level and living situation (living alone, yes/no) as additional control variables to rule out alternative explanations based on between-person (i.e., between-cluster) differences based on these variables. Cluster was tested as a moderator of the association between animal presence and PA/NA by adding the interaction term of animal presence cluster as a moderating independent variable. Specifically, cluster (1-5) was added as a moderator using dummy variables, testing the interaction effects between animal presence and cluster. No single reference category was used; we tested all possible pairwise comparisons between clusters.

**Table 1. Distribution of caregiver characteristics across clusters and in total sample.**

| | Cluster 1 | | Cluster 2 | | Cluster 3 | | Cluster 4 | | Cluster 5 | | Total | |
|---|---|---|---|---|---|---|---|---|---|---|---|---|
| | N | Col% | N | Col% | N | Col% | N | Col% | N | Col% | N | Col% |
| *Employment status*** | | | | | | | | | | | | |
| Unemployed (1) | 10 | 13.5 | 7 | 6.7 | 0 | 0 | 9 | 15.3 | 15 | 36.6 | 41 | 27.7 |
| Domestic Work (2) | 6 | 8.1 | 4 | 3.8 | 1 | 2.3 | 7 | 11,9 | 6 | 14.6 | 24 | 7.5 |
| School/Education (3) | 18 | 24.3 | 20 | 19.0 | 5 | 11.6 | 15 | 25.4 | 4 | 9.8 | 62 | 19.3 |
| Full-time Employed (> 32 hours) (4) | 32 | 43.2 | 52 | 49.5 | 26 | 60.5 | 14 | 23.7 | 8 | 19.5 | 132 | 41.0 |
| Part time Employed (≤ 32 hours) (5) | 8 | 10.8 | 22 | 21.0 | 11 | 25.6 | 14 | 23.7 | 8 | 19.5 | 63 | 19.6 |
| *Educational level** | | | | | | | | | | | | |
| Primary and Secondary – Short (1-6) | 14 | 18.9 | 18 | 17.0 | 3 | 7.0 | 15 | 25.4 | 12 | 29.3 | 62 | 19.2 |
| Tertiary – Long (7-9) | 60 | 81.0 | 88 | 83.0 | 40 | 93.0 | 44 | 74.6 | 29 | 70.7 | 261 | 80.8 |
| *Gender* | | | | | | | | | | | | |
| Women (0) | 54 | 72.00 | 72 | 67.9 | 27 | 62.8 | 34 | 62.8 | 34 | 82.9 | 221 | 68.2 |
| Men (1) | 21 | 28.00 | 34 | 32.1 | 16 | 37.2 | 25 | 37.2 | 7 | 17.1 | 103 | 31.8 |
| *Number of children* | | | | | | | | | | | | |
| 0 | 39 | 52.0 | 42 | 39.6 | 26 | 60.5 | 27 | 45.8 | 19 | 46.3 | 153 | 47.2 |
| 1 | 20 | 13.3 | 18 | 17.0 | 4 | 9.3 | 8 | 20.3 | 6 | 14.6 | 46 | 14.2 |
| 2 | 17 | 22.7 | 28 | 26.4 | 10 | 23.3 | 12 | 13.6 | 10 | 24.4 | 77 | 23.8 |
| 3 or more | 9 | 12 | 18 | 16.9 | 2 | 4.7 | 12 | 20.4 | 6 | 14.6 | 48 | 14.8 |
| *Animal species (both possible)* | | | | | | | | | | | | |
| Cats (at least 1) | 40 | 53.3 | 42 | 39.6 | 26 | 61.5 | 41 | 69.5 | 20 | 48.8 | 169 | 43.2 |
| Dogs (at least 1)** | 45 | 60.0 | 78 | 73.6 | 22 | 51.2 | 24 | 41.7 | 34 | 82.9 | 203 | 62.7 |
| *Marital status** | | | | | | | | | | | | |
| Single (1) | 15 | 20 | 12 | 11.3 | 2 | 4.7 | 10 | 16.9 | 5 | 12.2 | 44 | 13.6 |
| In a relationship (2) | 9 | 12 | 7 | 6.6 | 8 | 18.6 | 4 | 6.8 | 4 | 9.8 | 32 | 9.9 |
| Married or living together (3) | 50 | 66.7 | 82 | 77.4 | 33 | 76.7 | 43 | 72.9 | 25 | 61.0 | 233 | 71.9 |
| Divorced or Widowed (4-5) | 1 | 1.3 | 5 | 4.7 | 0 | 0 | 2 | 3.6 | 7 | 17.1 | 15 | 4.6 |
| *Living situation** | | | | | | | | | | | | |
| Alone (1 = yes) | 13 | 17.3 | 14 | 13.2 | 4 | 9.3 | 6 | 10.2 | 14 | 34.1 | 51 | 15.7 |
| Total | 75 | 100 | 106 | 100 | 43 | 100 | 59 | 100 | 41 | 100 | 324 | 100 |

*Notes.* \*$p < .05$ and \*\*$p < .01$ in Chi-square Test. $N$ = number of individuals, with $N_{persons}$ = 324 and $N_{observations}$ = 16,127. Col% = column percentage. There were missing data for the variables Employment status ($n_{missing}$ = 2) and Educational level ($n_{missing}$ = 1).

# Results

## Momentary analyses: Work state as moderator

Table 2 shows the results of the multilevel analyses to predict PA/NA. As can be seen in this table, 47.9% of the variance in positive affect and 55.8% in negative affect is due to individual variation. This supports our choice for analyses with variables both on the within-person (i.e., momentary analyses) and the between-person level (i.e., cluster analyses). Multilevel random regression analyses indicated a significant association between animal presence and PA (B = 0.155, $p < .001$, 95% CI = 0.103; 0.207; Step 1). This effect does not seem to differ between telework and non-work as no significant interaction effect between animal presence and work state was found (B = −0.087, $p = 0.273$, 95% CI = −0.242; 0.068; Step 1). In the model of NA, a significant interaction effect was found (B = .091, $p = 0.032$, 95% CI = 0.008; 0.173; Step 1).

**Table 2. Random coefficient modelling results of the momentary analysis of animal presence and affect, moderated by momentary (i.e., within-person level) work state (Step 1).**

| | Positive affect | | | Negative affect | | |
|---|---|---|---|---|---|---|
| | β | SE | p | β | SE | p |
| Animal presence (1 = yes) | 0.09 | 0.03 | .002 | −0.05 | 0.01 | .000 |
| Work state (1 = work at home) | −0.24 | 0.07 | .000 | 0.03 | 0.04 | .430 |
| Human presence (1 = yes) | −0.14 | 0.02 | .000 | 0.06 | 0.01 | .000 |
| Age (years) | 0.02 | 0.00 | .000 | 0.00 | 0.00 | .083 |
| Gender (1 = female) | 0.05 | 0.11 | .623 | 0.15 | 0.07 | .023 |
| Animal presence x Work state | −0.09 | 0.08 | .273 | 0.09 | 0.04 | .032 |
| Variance level 2 (person) | 0.47 (47.9%) | | | 0.53 (55.8%) | | |
| Variance level 1 (observation) | 0.52 (52.1%) | | | 0.42 (44.2%) | | |

Notes. SE = standard error. N = 324 persons and 16,127 observations.

Given this significant interaction, the association between animal presence and NA was tested stratified by each work state group. When off-work, a significant negative association was found between animal presence and NA (B = −.049, $p \leq .001$, 95% CI = −0.074; −0.023; Step 1). When teleworking, no association was found between animal presence and NA (B = .012, $p$ = 0.834, 95% CI = −0.097; 0.120; Step 1).

## Animal-work clusters

We identified five clusters with recognizable patterns of animal-work constellations over time (see Tables 3 and 4).

Individuals in Cluster 1 (C1: "Teleworking, off-work time with animal"; n = 75) had a relatively high amount of teleworking states—mostly together with an animal—and spent most of their non-working time with their animal. They were mainly female (72%) with a dog or a cat. Cluster 2 (C2: "Working, off-work time with animal"; n = 106) was the largest cluster with the highest number of missing states. Nevertheless, missing states largely occurred during working hours, and non-missing notifications mostly occurred during work or non-work time with an animal. Most of the respondents in this cluster were full-time employed. A large proportion of this cluster's sample included dog caregivers (74%) and individuals who were married or living together (77%). Individuals in Cluster 3 (C3: "Working, off-work time with and without animal"; n = 43) were also mainly working full-time and exhibited a pattern similar to that in C2, with less missing states and more non-working time states without an animal. This group had the highest proportion of childless individuals (61%) and 9% of this group lived alone. Cluster 4 (C4: "Teleworking, off-work time without animal"; n = 59) had a different dominant state to all the other groups, namely 'non-working without animal'. Cats were more common in this cluster (cat presence: 70%). Finally, Cluster 5 (C5: "Non-working; with animal", n = 41) included respondents that rarely indicated work as the current activity and spend most of time with their companion animals (dog presence: 82%). Fig 1 displays the proportions of the combined animal-work states for each measurement point in so-called "chronograms", organized by cluster. Fig 2 displays the index-plots organized by cluster of animal-work constellation.

## Momentary analyses; cluster as moderator

Multilevel random regression analyses showed that interaction terms between animal presence and certain clusters were significant, indicating that the association between animal presence and affect varies across cluster-pairs. Table 5 shows the coefficients for the interaction

**Table 3. Overview of the five distinguished clusters of animal-work constellation state trajectories.**

| Cluster | N | Col. % | Cluster Name | Dominant state |
|---|---|---|---|---|
| C1 | 75 | 23.1 | Teleworking, off-work time with animal | Non-work with animal |
| C2 | 105 | 32.7 | Working, off-work time with animal | Non-work with animal |
| C3 | 43 | 13.3 | Working, off-work time with & without animal | Non-work with animal |
| C4 | 59 | 18.2 | Teleworking, off-work time without animal | Non-work without animal |
| C5 | 41 | 12.7 | Non-working with animal | Non-work with animal |
| Total | 324 | 100.0 | | |

Notes. N = number of individuals, with $N_{persons}$ = 324 and $N_{observations}$ = 16,127. Col% = column percentage.

**Table 4. Distribution of respondent age and cumulative durations (i.e., number of beeps spent in state) in the animal-work constellations.**

| Variable | Cluster 1 | | Cluster 2 | | Cluster 3 | | Cluster 4 | | Cluster 5 | | Total | |
|---|---|---|---|---|---|---|---|---|---|---|---|---|
| | M | SD | M | SD | M | SD | M | SD | M | SD | M | SD |
| *Age* | 41.7 | 13.2 | 43.0 | 13.4 | 40.2 | 12.0 | 44.8 | 15.5 | 51.8 | 10.9 | 43.79 | 13.66 |
| *Cumulative duration* | | | | | | | | | | | | |
| Telework with animal (1) | 3.1 | 4.15 | 1.5 | 2.20 | 1.6 | 2.13 | 2.9 | 3.10 | 1.5 | 2.06 | 2.1 | 2.99 |
| Telework without animal (2) | 0.4 | 0.98 | 0.4 | 1.25 | 0.4 | 0.69 | 2.0 | 2.64 | 0.1 | 0.26 | 0.6 | 1.57 |
| At work with or without animal (3) | 2.5 | 3.00 | 5.5 | 4.86 | 13.2 | 4.31 | 2.4 | 3.14 | 2.0 | 3.20 | 4.8 | 5.30 |
| Non-work with animal (4) | 17.9 | 5.86 | 10.8 | 4.67 | 16.4 | 5.31 | 16.6 | 7.10 | 33.2 | 4.68 | 17.1 | 8.75 |
| Non-work without animal (5) | 7.5 | 4.88 | 5.4 | 3.54 | 9.6 | 3.81 | 17.9 | 7.64 | 6.8 | 3.66 | 8.9 | 6.59 |
| No information (6) | 18.5 | 4.06 | 26.4 | 4.32 | 8.9 | 4.37 | 8.3 | 4.37 | 6.4 | 3.40 | 16.5 | 9.12 |
| Total | 75 | 100 | 106 | 100 | 43 | 100 | 59 | 100 | 41 | 100 | 324 | 100 |

Notes. M = mean. SD = standard deviation. N = number of individuals, with $N_{persons}$ = 324 and $N_{observations}$ = 16,127. There were missing data for Age ($n_{missing}$ = 1).

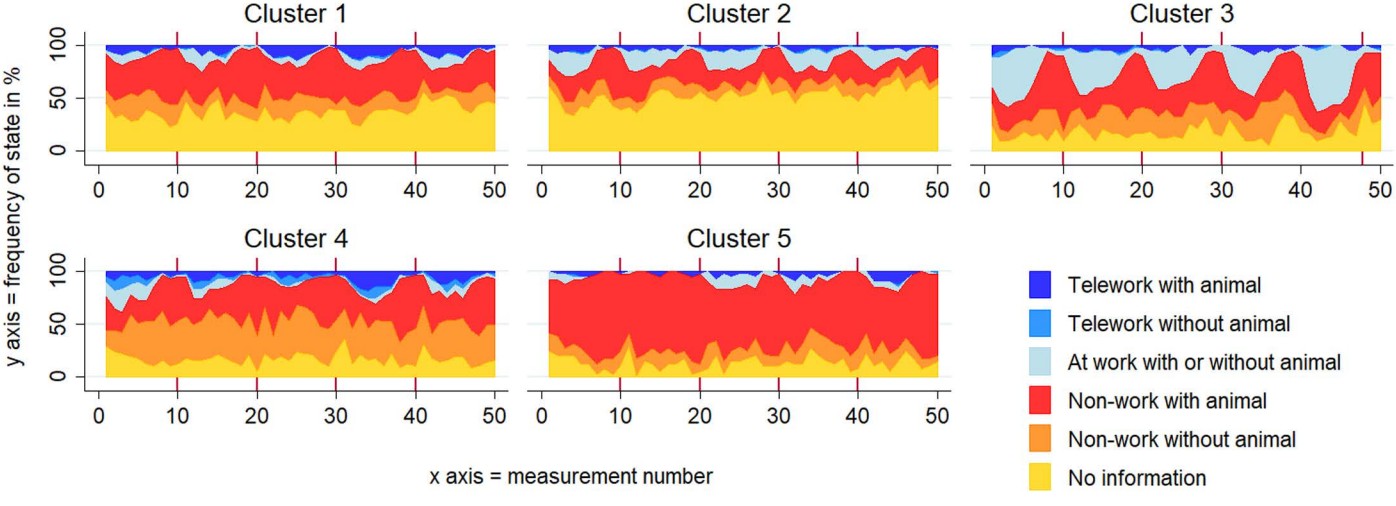

**Fig 1. Chronograms in the sequence analysis.** Chronograms by cluster of animal-work constellation ($N_{persons}$ = 324, $N_{observations}$ = 16,127).

terms. The association between animal presence and PA and NA in cluster 2 differed from several of the other clusters: In the model of PA, Cluster 4 differed from Cluster 2 (B = −0.164, p = 0.032, 95% CI = −0.314; −0.014). In the model of NA, Cluster 1 (B = 0.104, p = 0.005, 95% CI = 0.177; 0.032), Cluster 3 (B = 0.092, p = 0.019, 95% CI = 0.015; 0.169) and Cluster 4 (B = 0.101, p =.005, 95% CI = 0.030; 0.171) from Cluster 2.

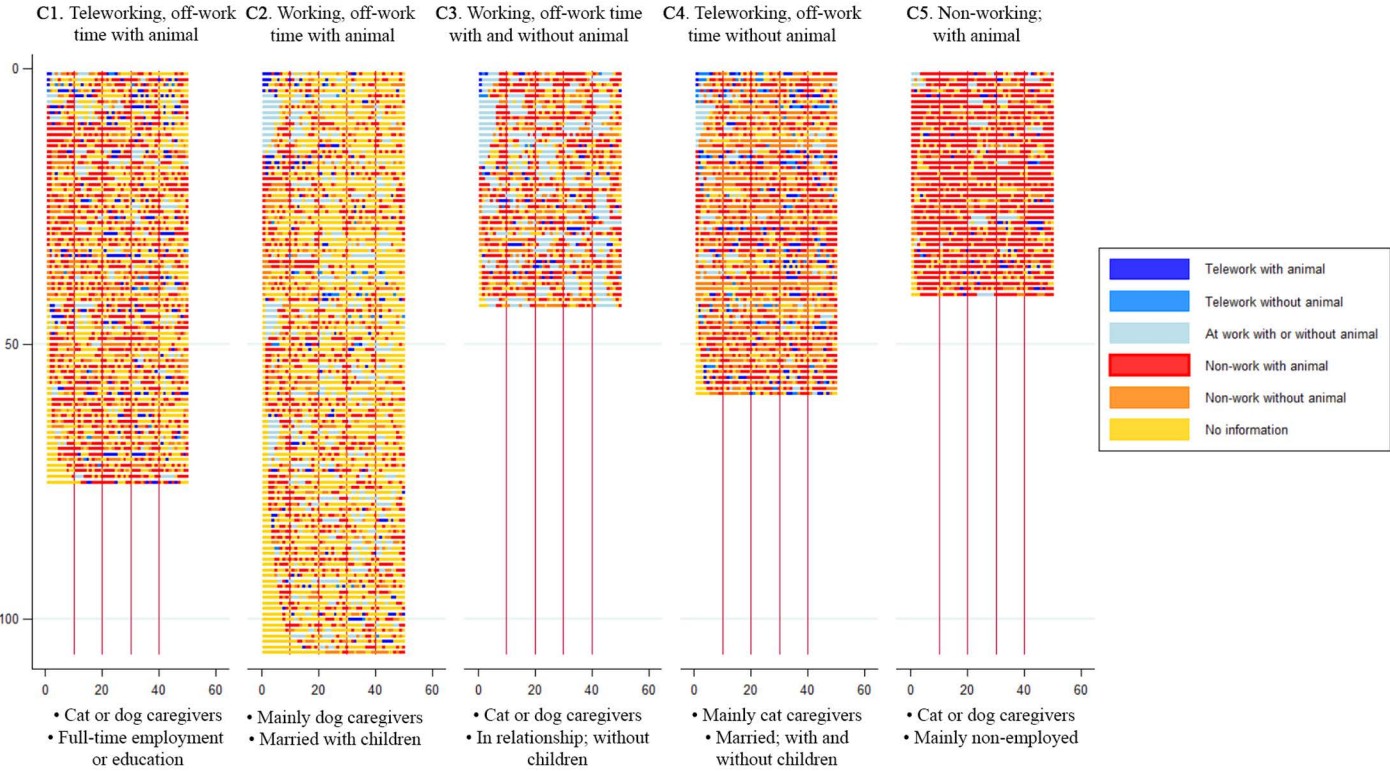

Note: Each horizontal line on the y-axis represents the sequence of measurements for one person in the cluster. The x-axis shows the measurement number.

**Fig 2. Index-plots in the sequence analysis.** Index-plots by cluster of animal-work constellation ($N_{persons}$ = 324, $N_{observations}$ = 16,127).

**Table 5. Interaction between pet-presence and cluster-pairs.**

| Cluster pair | PA | | NA | |
|---|---|---|---|---|
| | B | 95% CI | B | 95% CI |
| 2 vs 1 | .093 | −.058;.244 | −.104** | −.177; −.032 |
| 3 vs 1 | −.012 | −.182;.159 | −.012 | −.091;.066 |
| 4 vs 1 | −.071 | −.227;.085 | −.004 | −.076;.068 |
| 5 vs 1 | −.004 | −.186;.178 | −.026 | −.110;.059 |
| 3 vs 2 | −.104 | −.270;.061 | .092* | .015;.169 |
| 4 vs 2 | −.164* | −.314; −.014 | .101** | .030;.171 |
| 5 vs 2 | −.097 | −.274;.080 | .079 | −.004;.162 |
| 4 vs 3 | −.060 | −.229;.110 | .009 | −.068;.085 |
| 5 vs 3 | .007 | −.186;.201 | −.013 | −.102;.076 |
| 5 vs 4 | .067 | −.114;.248 | −.022 | −.104;.061 |

*Notes.* *$p$ < .05, **$p$ < .01, $N_{persons}$ = 324 and $N_{observations}$ = 16,127.

To further explore the association between animal presence and affect in each cluster, we stratified the data and tested the main effect of animal presence separately for each cluster. As can be seen in Table 6, results indicate that Cluster 2 shows the most prominent "pet-effect" illustrated by a significant association between animal presence and both PA and NA ($B$ = 0.246, $p$ < 0.001, 95% CI = 0.136; 0.357) and NA ($B$ = −0.146, $p$ < 0.001, 95% CI = −0.206;

Table 6.  Association between animal presence and caregiver affect (PA/NA) stratified by cluster.

| Cluster | | PA | | NA | |
|---|---|---|---|---|---|
| nr | N | B | 95% CI | B | 95% CI |
| 1 | 75 (2214) | .147* | .035;.260 | −.033 | −.083;.017 |
| 2 | 106 (2217) | .246** | .136;.357 | −.146*** | −.206; −.087 |
| 3 | 43 (1580) | .131* | .004;.258 | −.044 | −.102;.013 |
| 4 | 59 (2263) | .077 | −.017;.171 | −.041 | −.083;.001 |
| 5 | 41 (1739) | .145 | −.012;.302 | −.064 | −.148;.019 |

*Notes.* $*p <.05$, $**p <.01$, $***p <.001$. $N_{persons}$ = 324 and $N_{observations}$ = 16,127.

Analyses are corrected for Human presence, Age and Gender

−0.087). We also found a significant association with PA—but not NA—in Cluster 1 (PA: B = 0.147, $p < 0.05$, 95% CI = 0.035; 0.260) and Cluster 3 (PA: B = 0.131, $p < 0.05$, 95% CI = 0.004; 0.258). There were no significant effects in Cluster 4 or Cluster 5.

## Discussion

The main objective of this study was to gain a comprehensive understanding of how HAI across daily life contexts relate to human affect. We did this by assessing momentary associations between companion animal presence during work and non-work time and caregiver PA and NA (Aim 1), by exploring individual trajectories of animal-work constellation states over time and identifying between-person varying clusters of patterns of these trajectories (Aim 2), and by examining how these clusters moderate the momentary associations between animal presence and PA/NA (Aim 3). Data were collected in a convenience sample of 324 animal caregivers across 16,127 measurement occasions, allowing for respondent diversity concerning age, gender, employment status, educational level, number of children, marital status, living situation and animal species at home. Mixed modelling results showed that momentary animal presence was associated with higher PA regardless of whether caregivers were teleworking or not working, and with lower NA only when caregivers were not working at home. Next, sequence analysis on caregivers' individual trajectories of animal-work constellation states (i.e., working at work, teleworking with animal, teleworking without animal, not working with animal, not working without animal) revealed five clusters with distinct patterns in animal-work constellation state trajectories. We labeled Cluster 1 as "Teleworking, off-work time with animal" (C1), Cluster 2 as "Working, off-work time with animal" (C2), Cluster 3 as "Working, off-work time with and without animal" (C3), Cluster 4 as "Teleworking, off-work time without animal" (C4), and Cluster 5 as "Non-working with animal" (C5). Last, mixed modelling demonstrated that individuals with the "Working, off-work time with animal" pattern (C2) experienced the strongest "pet effect" (i.e., the beneficial association between animal presence and caregiver affect) compared to individuals exhibiting pattern C1 (NA), C3 (NA) and C4 (PA and NA), but did not significantly differ from individuals exhibiting pattern C5 (neither PA or NA). A visual overview of the clusters is presented in Fig 2.

Addressing our first aim, we contextualized the "pet effect" by identifying the moderating impact of the momentary work context (i.e., work state) on the momentary association between animal presence and NA—but not PA. Specifically, we found that a negative association between animal presence and NA was not present when caregivers were teleworking, and that there was a positive association between animal presence and PA irrespective of work state. This result nuances earlier findings that showed animal presence to buffer against NA [1,3,7] as it shows the lack of such a buffering effect in the teleworking context. Additionally,

it replicates other studies that showed animal presence to increase PA both in non-working contexts [1,3,7,25] and the teleworking context [26]. While our study did not assess explanatory mechanisms, explanations for the observed moderating impact of work state on relationship between animal presence and affect may have to do with different features related to the animal, the caregiver, and/or their interactions across non-working and teleworking contexts. For instance, animal behaviors that impact human mental health [44,64] have been found to differ across contexts, for instance depending on whether caregivers are nearby [65], whether caregivers spend increased time with their animal such as during the COVID-19 pandemic "lockdown" measures [66,67] and whether caregivers engage with their animal in particular activities [68]. Also studies on HAI in work contexts have shown differences between (tele) working and non-working contexts concerning animal behaviors [69] and caregiver behaviors related to animal caregiving [30,69–71]. Moreover, as species-specific features of HAI exist [40,68,72,73], scholars may benefit from conceptualizing their effects separately [74]. Likewise, scholars have called to include the specific behaviors of animals and their caregivers when nuancing the "pet effect" [55,75], as animal behavior is paramount for understanding effects of HAI, including potential disruptive or negative impacts [76,77]. Future studies may benefit from assessing animal and caregiver behaviors to understand context-dependent findings.

Next, addressing our second aim, we found meaningful differences in the unfolding of HAI states across contexts and time points, identifying five between-person varying clusters of individual trajectory patterns of animal-work constellations. Thus, clusters differed from one another based on the longitudinal pattern in which (tele)work and animal caregiving were combined. Specifically, C1 and C4 were characterized by teleworking caregivers, with caregivers in C4 spending their non-working time more often without their animal as compared to individuals from other clusters. Cats were more common as a companion animal in this cluster. Additionally, C2 and C3 were characterized by caregivers who spent longer time working at work compared to other clusters, with caregivers in C3 spending less time with their animal during non-working time compared to caregivers in C2. Additionally, C2 was characterized by a large proportion of dog caregivers (74%). Finally, C5 was characterized by caregivers who did not work. Hence, our findings show that caregivers can be categorized based on their patterns in HAI state trajectories based on animal presence (i.e., present or absent), working activity (i.e., working, not working) and location (i.e., at home or elsewhere). This approach, which takes into account both the animal and the caregiver across contexts and over time, aligns with a "one welfare" perspective that links animal with caregiver well-being [78,79], and addresses recent scholarly recommendations to assess HAI at an interactional level taking into account features of the animal, the caregiver, and their prevailing complex environments [15,27,54]. Future studies could further enhance this approach by incorporating greater complexity concerning additional aspects related to the longitudinal organization of features involving the animal, caregiver or context. This may include the behaviors of both animal and caregiver, as well as specific characteristics of the HAI, such as the intensity of animal-caregiver interactions, within individual HAI trajectories across various contexts and over time [55,70,80,81]. Thus, in addition to considering the moderating effects of human or animal behaviors and HAI characteristics (e.g., interaction features), it is crucial to examine the longitudinal organization of individual behavioral and interaction patterns of both humans and animals involved.

Last, in addressing our third aim, we contextualized both the "pet effect" and the momentary analyses addressed in our first aim by demonstrating how between-person variations in the trajectories of animal-work constellation states moderate the momentary "pet effect". Specifically, individuals exhibiting the pattern in Cluster 2 experienced the strongest "pet effect" compared to individuals from Cluster 1 (NA), Cluster 3 (NA) and Cluster 4 (PA

and NA), but did not significantly differ from individuals exhibiting the pattern of Cluster 5 (NA). Importantly, this moderating effect was not alternatively explained by mere between-person differences, as we added age, employment, education and living situation as covariates in our analyses. Our findings highlight the importance of considering the longitudinal complexity of HAI over contexts (i.e., clusters based on patterns of state trajectories) when understanding the momentary "pet effect" and when understanding our observed moderating impact of work state in our first aim. While our study did not ascertain *why* the specific HAI pattern of individuals in C2 moderated the momentary association between animal presence and affect, we suggest two possible explanations. First, individuals in C2 may experience their time together with their animal as more worthy since the interactions with the animals "compensate" for the daily life hassles often associated with working time—especially when working outside of the home [30,82,83]. Second, making up for missed time spent together with an animal during the workday may increase caregivers' perceived intensity of HAI, facilitating HAI's beneficial psychophysiological effects [84] and "mindful" impact [29,85], improving affect.

Overall, our findings demonstrate that a high-resolution approach, combining EMA with sequence analysis, effectively evaluates the significance of between-person variability in state trajectories (i.e., between-person patterns) related to living with animals in daily life contexts. This approach captures the complex dynamics of HAI and their interplay with activities like work. While traditional low-resolution HAI measures, which focus on momentary animal presence without considering longitudinal patterns across contexts, provide valuable insights, they are limited in assessing the complexity of these interactions. Therefore, high-resolution approaches like sequence analysis complement these traditional methods by offering a nuanced perspective on the longitudinal organization of states [41], such as HAI. Specifically, our momentary analyses show detailed phenomenological insights into momentary HAI and associated affective states, while our sequence analysis uncovers their longitudinal complexity across contexts. Importantly, our findings show that this high-resolution approach contextualizes the "pet effect" by highlighting between-person variability in HAI state trajectories (i.e., patterns) as an important moderator of this effect.

## Limitations

Some issues need to be accounted for when interpreting our results. First, all measures were self-reported, which could potentially increase common-method bias [86–88]. Additionally, we considered the temporal sequences of our variables in our sequence analysis to reduce the risks of common method bias. Nevertheless, this concern is mitigated by the presence of our observed interaction effects, as common method variance tends to decrease the likelihood of detecting such effects [88]. Future research on the "pet effect" and the longitudinal dynamics of HAI across different contexts could benefit from incorporating both self-reported data and "objective" well-being measures. This might include physiological measurement via wearables [89] and HAI video-recordings [90].

A second limitation concerns study representativity, especially concerning respondent gender and educational level. Whereas convenience samples are considered to be suitable for elaborated research designs such as EMA studies [56,57], in our study, there was a female response rate of 72%, and 80% of respondents had an educational level of the long type. Therefore, the generalizability of our findings towards other genders and short-type educational levels warrants caution.

Third, our study encountered numerous instances of missing data, a common issue in EMA research. If these missing data are systematic, they should be addressed appropriately. Potentially, the missing states might correlate with respondents' HAI and working location

and activities. Future research should consider strategies to reduce missing data, such as providing reminders or offering incentives for completing entries.

Fourth, our study does not permit causal claims as there is potential for reversed causality. However, our study design includes observational measurements throughout the day and considers longitudinal complexity across contexts, which enhances the ability to infer directionality of the results [91]. Our approach aligns with three key recommendations for improving causal inference from observational data [92]. Specifically, our design 1) repeatedly assesses the same constructs over time; 2) considers the temporal sequence of effects between constructs; and 3) employs an analytical strategy that differentiates within- from between-person effects [92].

Last, because we utilized secondary data, the data in our sample was originally collected for different research purposes and without prior knowledge of our specific research aims. Consequently, the study protocol was not optimized for our objectives. Notably, data collection did not span a complete (working) week and included non-working days, leading to a skewed dataset with a predominance of 'non-working with animal' states. Future studies on this topic should ideally collect data over a full week to achieve more consistent sequences that are easier to compare. Additionally, some data were collected during the COVID-19 pandemic. Nevertheless, since data were gathered when telework measures were in place but not mandatory (i.e., not daily required work-from-home), our findings offer insights relevant to current and future contexts where many organizations adopt hybrid working models that include remote work for knowledge workers. Lastly, we did not assess the duration of time animals had resided with their caregivers, which might obscure variance attributable to caregivers' familiarity with their animals (e.g., duration or strength of the animal-caregiver bond [21]). Since the length of cohabitation could influence HAI outcomes, future research may benefit from replicating our findings in samples where caregivers and animals have cohabitated for a defined period. However, given our focus on the longitudinal patterns of HAI across contexts, it is likely that the effects of cohabitation duration are already embedded in these patterns. This strengthens the robustness of our approach to examining HAI and its longitudinal impacts across diverse contexts.

## Author contributions

**Conceptualization:** Joni Delanoeije, Miriam Engels, Mayke Janssens.

**Data curation:** Mayke Janssens.

**Formal analysis:** Joni Delanoeije, Miriam Engels, Mayke Janssens.

**Funding acquisition:** Joni Delanoeije.

**Investigation:** Joni Delanoeije, Miriam Engels, Mayke Janssens.

**Methodology:** Joni Delanoeije, Miriam Engels, Mayke Janssens.

**Project administration:** Joni Delanoeije, Mayke Janssens.

**Visualization:** Miriam Engels.

**Writing – original draft:** Joni Delanoeije, Miriam Engels, Mayke Janssens.

**Writing – review & editing:** Joni Delanoeije, Miriam Engels, Mayke Janssens.

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
