## [Decision Letter · Decision Letter 0]

7 Nov 2024

Dear Dr. Delanoeije,

Thank you for submitting your manuscript to PLOS ONE. After careful consideration, we feel that it has merit but does not fully meet PLOS ONE’s publication criteria as it currently stands. Therefore, we invite you to submit a revised version of the manuscript that addresses the points raised during the review process.

The Manuscript is well written, but the Reviewers final requirements will entitle the final version with the elegance and scientific soundness the Journal is exhibiting. 

We look forward to receiving your revised manuscript.

Kind regards,

Ioana Gutu, Postdoctoral

Academic Editor

PLOS ONE

Journal Requirements:

2. In the online submission form, you indicated that data cannot be shared publicly because of ethical considerations related to the study respondents: the data cannot be made publicly available as per the ethical approval granted for the original studies. Data are available from the Research Ethics Committee (cETO) of the Open University of the Netherlands (OU; U2016/00165/CBO and U2022/08386), (contact via mayke.janssens@ou.nl) for researchers who meet the criteria for access to confidential data.

Materials and quantitative analysis methods (including code) that support the findings of this study are available from the corresponding author upon request. 

Reviewers' comments:

Reviewer's Responses to Questions

**Comments to the Author**

1. Is the manuscript technically sound, and do the data support the conclusions?

Reviewer #1: Partly

Reviewer #2: Partly

2. Has the statistical analysis been performed appropriately and rigorously?

Reviewer #1: Yes

Reviewer #2: Yes

3. Have the authors made all data underlying the findings in their manuscript fully available?

Reviewer #1: No

Reviewer #2: No

4. Is the manuscript presented in an intelligible fashion and written in standard English?

Reviewer #1: Yes

Reviewer #2: Yes

Reviewer #1: Thanks for your article which shows a different dimension to human animal interaction. I wondered if you have heard of the below articles or have any thoughts as to how this may modify your discussion and outcome.

Companion animals and child development outcomes: longitudinal and cross-sectional analysis of a UK birth cohort study | BMC Pediatrics | Full Text (biomedcentral.com)

The Complexity of the Human–Animal Bond: Empathy, Attachment and Anthropomorphism in Human–Animal Relationships and Animal Hoarding - PMC (nih.gov)

Working with animals: Implications for employees’ compassion, awe, prosocial behavior, and task performance - Yam - 2023 - Personnel Psychology - Wiley Online Library

Also, whilst I realise that this is a complex topic as well as a complex methodology to navigate around is there a way to optimise pictorials ie showing a diagram of the clusters with linked summary at the end or prior tot he discussion to make it easier to follow the assumptions and the outlined associations inferred ?

Otherwise well written despite its complexity.

Reviewer #2: I have read the manuscript thoroughly and I appreciate the effort. The importance of Human-animal interaction with caregivers cannot be overemphasized and this manuscript lay it bear. I will implore the authors to consider stating years or months the respondent had been with the dog or cat as an inclusion criteria or part of their analysis.

Also, the conclusion was not clearly written so that can be addressed.

**Do you want your identity to be public for this peer review?** For information about this choice, including consent withdrawal, please see our Privacy Policy

Reviewer #1: **Yes: ** Olubunmi Arogunmati

Reviewer #2: No

---

## [Author Response · Author response to Decision Letter 0]

23 Dec 2024

Data cannot be shared publicly because of ethical considerations related to the study respondents. We did not explicitly seek participants' consent to publicly share their data, as this was not anticipated at the time of ethical review. Additionally, the data collected involve sensitive, personal information obtained through experience sampling methodology (ESM), in which participants shared details about their daily lives. Given the nature of this data, there are heightened privacy considerations.

The policy of the Open University of the Netherlands is to take a cautious approach regarding public data sharing in cases involving such sensitive information. This approach aligns with our commitment to protecting participant confidentiality in studies where data capture highly personal aspects of daily experiences.

In light of these considerations, we respectfully request an exemption from the requirement to publicly deposit the data, as we prioritize adherence to the ethical standards set for this study. Access to the data however may be considered for qualified researchers who meet specific criteria, subject to evaluation by the Research Ethics Committee of the Open University (cETO; https://www.ou.nl/en/research-ethics-committee-ceto; OU; U2016/00165/CBO and U2022/08386). The contact person for data availability request is Professor Dr. Nele Jacobs, Department Head, Life Span Psychology, Open University, the Netherlands; e-mail address: nele.jacobs@ou.nl.

---

## [Decision Letter · Decision Letter 1]

26 Jan 2025

“Pet effect” patterns: Dynamics of animal presence and caregiver affect across (tele)work and non-work contexts

PONE-D-24-27055R1

Dear Dr. Joni Delanoeije,

We’re pleased to inform you that your manuscript has been judged scientifically suitable for publication and will be formally accepted for publication once it meets all outstanding technical requirements.

Kind regards,

Ioana Gutu, Postdoctoral

Academic Editor

PLOS ONE

Additional Editor Comments (optional):

**The manuscript is accepted under the following reserve**

As one of the Reviewers pointed out, please can you kindly ensure that headings do not fall in awkward points ie end of page as when reading Materials and Methods pg 7 line 310,Data Analysis pg 11 line 310, subsequent headings positioned after this will be impacted on by placement of figures.

Also, please could you clarify with authors about line 362/3 page 14 ie Dudahart and Calinsky measures based on within- and between-errors

363 from clusters with no clear results regarding the quality of the solutionDuda Hart Index(DHI) is currently outlined ie Duda-Hart, D -Index or Duda Index (https://permetrics.readthedocs.io/en/latest/pages/clustering/DHI.html) and Calinsky measure as Calinski -Harabasz Index ( in R) (https://stats.stackexchange.com/questions/52838/what-is-an-acceptable-value-of-the-calinski-harabasz-ch-criterion)

Reviewers' comments:

Reviewer's Responses to Questions

**Comments to the Author**

Reviewer #1: All comments have been addressed

Reviewer #2: All comments have been addressed

2. Is the manuscript technically sound, and do the data support the conclusions?

Reviewer #1: Yes

Reviewer #2: Yes

3. Has the statistical analysis been performed appropriately and rigorously?

Reviewer #1: Yes

Reviewer #2: Yes

4. Have the authors made all data underlying the findings in their manuscript fully available?

Reviewer #1: No

Reviewer #2: No

5. Is the manuscript presented in an intelligible fashion and written in standard English?

Reviewer #1: Yes

Reviewer #2: Yes

Reviewer #1: Thank you for this article and the revisions for which I am grateful.

Also, please could you clarify with publishers/editor about line 362/3 page 14 ie Dudahart and Calinsky measures based on within- and between-errors

363 from clusters with no clear results regarding the quality of the solutionDuda Hart Index(DHI) is currently outlined ie Duda-Hart, D -Index or Duda Index (https://permetrics.readthedocs.io/en/latest/pages/clustering/DHI.html) and Calinsky measure as Calinski -Harabasz Index ( in R) (https://stats.stackexchange.com/questions/52838/what-is-an-acceptable-value-of-the-calinski-harabasz-ch-criterion)?

All the best.

Reviewer #2: (No Response)

**Do you want your identity to be public for this peer review?** For information about this choice, including consent withdrawal, please see our Privacy Policy

Reviewer #1: No

Reviewer #2: No

---

## [Editor Report · Acceptance letter]

PONE-D-24-27055R1

PLOS ONE

Dear Dr. Delanoeije,

I'm pleased to inform you that your manuscript has been deemed suitable for publication in PLOS ONE. Congratulations! Your manuscript is now being handed over to our production team.

Kind regards,

on behalf of

Dr. Ioana Gutu

Academic Editor

PLOS ONE